# A Review of Potassium-Rich Crop Residues Used as Organic Matter Amendments in Tree Crop Agroecosystems

Ellie M. Andrews *, Sire Kassama, Evie E. Smith, Patrick H. Brown and Sat Darshan S. Khalsa

Department of Plant Sciences, College of Agricultural and Environmental Sciences, University of California Davis, Davis, CA 95616, USA; skassama@ucdavis.edu (S.K.); evsmith@ucanr.edu (E.E.S.); phbrown@ucdavis.edu (P.H.B.); sdskhalsa@ucdavis.edu (S.D.S.K.)
* Correspondence: hemandrews@ucdavis.edu

**Abstract:** Ecosystem-based approaches to nutrient management are needed to satisfy crop nutrient requirements while minimizing environmental impacts of fertilizer use. Applying crop residues as soil amendments can provide essential crop nutrient inputs from organic sources while improving nutrient retention, soil health, water conservation, and crop performance. Tree crop hulls, husks, and shells have been found to contain high concentrations of potassium across species including almond, cacao, coffee, pecan, and hazelnut. The objective of this review is to characterize organic sources of potassium focusing on lignocellulosic pericarps and discuss reported effects of surface application on potassium cycling, water dynamics, soil functionality, and crop yield. Research indicates potassium ions solubilize readily from plant material into soil solution due to potassium's high mobility as a predominately unbound monatomic cation in plant tissues. Studies evaluating tree crop nutshells, field crop residues, and forest ecosystem litter layers indicate this process of potassium release is driven primarily by water and is not strongly limited by decomposition. Research suggests orchard floor management practices can be tailored to maximize the soil and plant benefits provided by this practice. Contextual factors influencing practice adoption and areas for future study are discussed.

**Keywords:** potassium; soil fertility; water; nutrients; organic matter amendments; yield; tree crops; soil health; agroecosystem management

## 1. Introduction

Sustainable nutrient management is critical if we are to satisfy crop nutrient requirements while minimizing impacts on human and environmental health [1,2]. As agricultural wastes increase worldwide, regional crop residues used as soil amendments can provide crop nutrient inputs while supporting nutrient cycling and retention, soil and water conservation, and soil biology [3]. Recycling crop residues provides an efficient strategy for reusing co-products which otherwise may pose financial and ecological waste management hurdles [4–6]. This practice has the potential to reduce residue burning and offsite disposal, thereby lessening air pollution and groundwater contamination [7–9]. Increased interest in agricultural resource use efficiency at the local scale has directed attention toward recycling crop residues and mulches as soil amendments due to relative availability, economic accessibility, and the potential to lower carbon footprints of crop production chains [10–13]. At the crop system level, residues can be strategically used as surface amendments to improve soil health [12,14], water and nutrient use efficiency [15], and yields, particularly under water or nutrient limited conditions [15–17]. This practice offers critical ecosystem services that can protect agricultural soils and enhance regional water and air quality [18]. In contrast, relying solely on inorganic fertilizer can lead to soil organic matter (SOM) loss, erosion, poor drainage, nutrient leaching, and regional water contamination issues [1,18,19]. As a critical indicator of soil health [19], the SOM pool consists of all organic detritus at various stages of decomposition. Nutrient management practices that harness agricultural ecosystem processes can strategically integrate mineral

and organic nutrient sources while building nutrient reservoirs [2]. Provisioning ecological services through optimized soil management is a central component of sustainable nutrient management in tree crop systems [20].

While nitrogen and phosphorus cycling in agricultural and forest ecosystems have been extensively investigated, studies focused on potassium (K) cycling and retention are relatively scarce despite the central role of K in plant function [21,22]. Many tree crops have high K demand particularly when fruits and nuts are ripening as sufficient K helps ensure fruit production and high quality [23–25]. In intensive tree crop systems where K supply is substantial, K accumulation in the fruit components may be especially high. For instance, Muhammad et al. (2015) determined 75 kg of K is removed in fruit (kernel, shell, and hull) per 1000 kg of kernel almond yield, exceeding N removal of 68 kg per metric ton [26]. Potassium fertilizer used to fulfill this demand can be prohibitively expensive in some regions, which can impose limitations on crop productivity [21]. Closing on-farm K cycles and reducing reliance on conventional K fertilizers can provide a strategy to improve crop system sustainability. Studies indicate K from crop residues can substitute for a portion of K fertilizer to fulfill crop requirements, reduce fertilizer expenses, and promote soil and crop benefits [27–31]. This practice can enhance crop system K cycling, reduce nutrient export, and moderate chronic depletion of soil K in intensive agricultural systems where access to conventional K fertilizer may be limited [7,32–35]. Organic-sourced K from recycled crop residues is a critical component of sustainable K management [36]. Across different crop systems, around 70–80% of K in harvested crop biomass could be retained on site if crop residues such as nutshells and straw remain in the field [7,33,37]. Nutrient cycling from tree crop leaf fall and alleyway biomass can be harnessed as valuable nutrient resources already present in orchard ecosystems [38]. Plant residue retention offers a strategy to limit K losses from agroecosystems while improving soil health and plant functioning.

For instance, in regions where nut crops are common, nutshells can be highly available and inexpensive nutrient-rich residues typically sold as mulches, feed, or bedding for livestock, or otherwise burned or discarded [39]. A growing body of research suggests that applying nutshells as amendments can help retain valuable nutrients on site, improve agroecosystem functioning, and create favorable plant growth conditions to improve crop performance. Nutshells are lignocellulosic materials that contain around 1 to 7% K (Table 1) and often have relatively high carbon (C) to nitrogen (N) ratios (C:N > 30:1). Amendments with high C content can increase SOM, soil quality, and yield over time [7,11,40]. Further research is needed to characterize potential cycles of N bioavailability under high C:N amendments, which often depend on a variety of contextual factors. However, N and K cycles are driven by different processes, and microbial communities do not substantially immobilize K [41]. Nutshells are a source of readily available organic K that solubilizes easily under water application as a plant-available monovalent cation ($K^+$). Prior research demonstrated that residue K solubilization is driven primarily by water, occurs rapidly, and is not strongly limited by C:N or residue decomposition rate [30,38,42–46]. A better understanding of amendment nutrient release rates and timing will assist researchers and growers in utilizing recycled organic matter to help meet crop nutrient demands [13].

This review examines current knowledge of the utilization of crop residues as organic matter amendments to supply K and enhance crop system function, with a focus on nutshells applied in orchards. Processes of K solubilization, soil availability, plant uptake, crop responses, and productivity are explored from a whole-system perspective. This review aims to survey current understanding of these dynamics and provide a springboard for future research investigating this practice. While K provided through crop residues is the central focus, this review additionally describes associated improvements in soil health and crop functioning and potential areas for future research.

## 2. Residue Decomposition, Nitrogen Cycling, and Potassium Solubilization

Crop residues provide carbon inputs that can slowly build SOM over time. Decomposition processes are controlled by biological activity and contextual variables such as

rainfall, irrigation, temperature, soil management, existing SOM levels, and microbial community composition [41,44,45,47]. Carbon transformation processes and products are functions of a specific soil ecosystem and are influenced by these environmental factors across spatial and temporal scales [48,49]. Decomposition occurs along a continuum as soil microbial, chemical, and physical processes produce a variety of organic and inorganic compounds from plant residues [49,50]. These processes are driven by cycles of soil microbial communities and associated physiology and enzymes [50]. For instance, the degradation of almond shell lignin produced a progressive release of C compounds, increasing the variety of C sources that could be further metabolized [51]. In general, lower temperatures, lower moisture levels, and higher C:Ns tend to be linked with slower decomposition rates [30,45]. While initial decomposition rates may correlate with chemical components indices such as N or lignin content [48], biological and physical controls of C flows between pools strongly influence soil organic carbon (SOC) dynamics and persistence [52]. These drivers and controls of decomposition should be considered when designing residue management approaches. For instance, in dry environments high C:Ns amendments such as almond shells tend to decompose slowly on the soil surface and thus have been recommended for tree crops in water-limited regions to provide a stable soil surface barrier layer and to slow nutrient leaching [10,16,33]. On the other hand, pericarps with lower C:N such as cacao husks in tropical environments tend to decompose more quickly and have been recommended as an organic nutrient source in depleted soils [53,54]. This highlights the importance of considering the effects of a site's climate, soil characteristics, management history, and residue composition on decomposition when designing residue management approaches to meet specific agroecosystem goals.

Studies evaluating the use of tree crop nutshells as amendments commonly note that the relatively high C:Ns (>30:1) can lead to N immobilization by microorganisms and potentially reduce crop N status. Organic-sourced N availability may be challenging to predict, considering the environmental and management-related variables controlling N cycling. Nitrogen immobilization under high C:N amendments may require increased short-term N fertilizer rates to ensure sufficient crop N. This was found to be true with compost–wood mixtures (30 and 60 Mg ha$^{-1}$) [55] and bark mulch (10 cm depth) [40]. However, other studies indicated that N immobilization may not necessarily impact crop N status or, in other cases, may improve crop N. For instance, Granatstein et al. (2014) did not find significant changes in apple leaf N under 10 cm thick wood chip mulch [56]. TerAvest et al. (2011) found that wood chip mulch resulted in slight N immobilization but led to high yield and tree growth [57]. Similarly, no effects of N immobilization on crop N status were found with macadamia nutshells (5 cm depth) [58] and pecan wood chip amendments (18,000 kg ha$^{-1}$) [59]. On the other hand, increased leaf N has been found in cashew seedlings under coffee husk residues [60] and in apple tree wood chip mulch [57]. Some sources have advised against high C:N amendments in young orchards without supplemental N fertilizer, while other studies indicated high C:N amendments could lead to vigorous young tree growth and high yields [57,61]. While potential N deficiency is a major concern associated with amendments in organic tree crop production, organic nutrient management practices are needed to moderate chemical N inputs and N losses while synchronizing N availability with crop demand [40,61]. A holistic framework is needed to assess processes driving the plant–microbe–mineral regulation of N bioavailability cycles [52,62,63] and develop more nuanced organic matter management strategies to recouple carbon and N cycles across diverse agroecosystems, soils, and climates. While N cycling in crop systems is influenced by many variables, the dynamics controlling organic-sourced K availability appear more simplistic and potentially predictable.

Potassium is the most abundant cation in plant cells [21,64]. Typically, nutrient content in crop residues is influenced by nutrient and water management, soil characteristics, crop-specific nutrient demands, and phenological stage at harvest as plant K dynamics change over the season [31,36,65,66]. A substantial fraction of annual K uptake can accumulate in tree fruits during development. For instance, Muhammad et al. (2015) found that

around 91% of whole plant K accumulation in almonds was allocated to fruit tissues [26]. Additionally, nutrient concentrations can vary substantially across regions as shown in cacao husk [65,67]. Post-harvest processing can influence residue K concentrations; for example, composting can substantially increase K content [68]. Table 1 shows ranges of K values in nutshells across crop species. Based on available literature, Table 1 demonstrates cacao, almond, coffee, hazelnut, and pecan pericarps often contain substantial K levels. Supplementary Diagram 1 illustrates examples of tree crop pericarp materials. Post-processing residues from olive and grape contain notable amounts of K as well, as do a wide variety of row crop residues. Studies have found nutshell biochar can supply K [69,70]; however, this review focuses on residue use without combustion to evaluate practices that minimize environmental pollution. Organic matter amendment nutrient levels are commonly evaluated through mass spectrometry techniques.

**Table 1.** Estimated percent potassium (% K) in residues from a variety of tree crops, other permanent crops, and row crops. Most sources did not report standard deviations.

| Source Crop | Material | Estimated %K | Region and Reference |
|---|---|---|---|
| **Residues and biomass from tree crops and other permanent crops** | | | |
| Almond | hull | 3.2 | Murcia, Spain; Valverde et al., 2013 Table 2 |
| Almond | shell | 0.5 | Murcia, Spain; Valverde et al., 2013 Table 2 |
| Almond | hull | 3.3 | California, U.S.A.; Atkas et al., 2015 Table 6 |
| Almond | shell | 1.7 | California, U.S.A.; Atkas et al., 2015 Table 6 |
| Bhimal (Grewia) | leaf litter | 2.55 | Himachal Pradesh, India; Verma et al., 2012 Table 3 |
| Cacao | husk | 2.89 | Cote d'Ivoire and the Netherlands; Hougni et al., 2021 Table 1 |
| Cacao | husk | 3.18 | Review; Campos-Vega et al., 2018 Table 2 |
| Cacao | husk | 3.73 | Pingtung, Taiwan; Tsai and Huang et al., 2018 Table 2 |
| Cacao | husk | 2.8–3.8 | Review; Lu et al., 2018 Table 1 |
| Cacao | husk | 3.78 | Akure, Nigeria; Agele et al., 2008 Table 1 |
| Cacao | husk | 1.6 | Cote d'Ivoire; Kone et al., 2021 Discussion |
| Cacao | husk | 3.77–7.69 | Review; Hartemink et al., 2005 Table 3 |
| Coffee | husk | 4.57 | Brazil; Carmo et al., 2016 Table 2 |
| Coffee | pulp husk mixture | 2.49 | Kinshasha, Congo; Kasongo et al., 2011 Table 2 |
| Coffee | pulp | 3.89 | Sao Paulo, Brazil; Zoca et al., 2014 Table 2 |
| Coffee | husk | 3.47 | Sao Paulo, Brazil; Zoca et al., 2014 Table 2 |
| Gliricidia | leaves | 2.65 | Malaysia; Zahara et al. 1999 Results |
| Grape | stalk | 3.0 | Spain; Bustamante et al., 2008 Table 1 |
| Grape | pomace | 2.42 | Spain; Bustamante et al., 2008 Table 1 |
| Grape | wine lee | 7.28 | Spain; Bustamante et al., 2008 Table 1 |
| Hazelnut | husk | 4.29 | Turkey; Kizilkaya et al., 2008 Results |
| Olive | pruned material | 0.56 | Review; Zipori et al., 2020 Table 1 |
| Olive | leaf | 0.69–1.19 | Review; Zipori et al., 2020 Table 2 |
| Olive | olive mill waste | 2.10 | Italy; Altieri et al., 2008 Table 1 |
| Olive | olive mill waste | 2.6 | Spain; Cayuela et al., 2004 Table 1 |
| Pecan | husk | 3.47 | New Mexico, U.S.A.; Idowu et al., 2017 Table 2 |
| Pecan | shell | 0.15 | New Mexico, U.S.A.; Idowu et al., 2017 Table 2 |
| Poplar | leaf litter | 1.24 | Himachal Pradesh; Verma et al., 2012 Table 3 |
| **Residues from row crops** | | | |
| Alfalfa | mulch | 2.2 | B.C., Canada; Neilsen et al., 2003 Table 1 |
| Black oat | green manure | 2.86 | Brazil; Franchini et al., 2003 Table 1 elongation stage |
| Black oat | residue | 4.22 | Brazil; Miyazawa et al., 2002 Table 1 |
| Maize | straw | 1.48 | Heilongjiang, China; Dong et al., 2019 Materials & Methods |
| Maize | residue | 1.53–1.69 | New Delhi, India; Madar et al., 2020 Table 1 |
| Radish | green manure | 3.85 | Brazil; Franchini et al., 2003 Table 1 vegetative stage |
| Rice | straw | 2.7 | Punjab, India; Yadav et al., 2019 Materials and Methods |
| Rice | residue | 2.1 | Punjab, India; Yadvinder-Singh et al., 2010 Methods and Materials |
| Rice | straw | 2.19 | Wuhan, China; Li et al., 2014 Materials and Methods |
| Rye | residue | 2.76 | Brazil; Miyazawa et al., 2002 Table 1 |
| Ryegrass cover crop | residue | 5.15 | Italy; Tagliavini et al., 2007 Table 1 |
| Soybean | straw | 1.05 | Heilongjiang, China; Dong et al., 2019 Materials & Methods |
| Sunflower | residue | 2.77 | Spain; Rodriguez-Lizana et al., 2010 Results |
| Wheat | straw | 1.22–1.91 | Jiangsu, China; Sui et al., 2014 Table 2 |
| Wheat | straw | 2.26–2.60 | New Delhi, India; Madar et al., 2020 Table 1 |
| Wheat | straw | 3.78 | Shaanxi, China; Wei et al., 2015 Table 1 |

In plant cells, potassium ions ($K^+$) function as highly mobile osmolytes that form weak complexes and remain readily exchangeable [64]. Plant residue K is predominantly present in soluble form in cell cytosol [21,42,45,71]. Numerous studies have demonstrated that K is rapidly released from plant residues through water extraction [30,43,45,71]. This process is typically characterized by extremely high release rates after initial water application followed by a slower release stage [30,38,45,46,72,73]. The small size and high mobility of K in plant cell solution enables solubilization from plant residues at rapid release rates largely independent of decomposition rates [42,44–46]. As a result, K release rates from plant residues tend to be dramatically faster than mass decomposition rates [45,71], although decomposition rate may influence K release in later release stages to a lesser degree. Compared to other macronutrients, K often demonstrates the most rapid release rates across a variety of crop residues [30,43,72–74], composts [75], and leaf litter [38,44,76,77].

The quantity and frequency of applied water determines the rate and total amount of K solubilization from plant material. For instance, Hougni et al. (2021) found K released rapidly from cacao pod husks at rates that varied as a function of rainfall frequency and quantity [43]. A study comparing straw residues found 10–20 mm of precipitation led to the greatest K release while less than 5 mm of precipitation did not release significant amounts of K [71]. Maize and soybean residues released around 95% of total K contents under 275 mm precipitation over two months [30]. When inundated with water, rice straw residues have been shown to release 90% total K after three days [45]. Zahara et al. (1999) found green manures released 95% of total K during the rapid initial release phase and 99.99% was released after 70 days under 689 mm of total rainfall [73]. Considering K solubilization is driven by water and tree K uptake occurs through water uptake, strategically timed water applications during periods of crop demand could be used to supply K from residues in a fashion similar to inorganic fertilizers. As many prior studies on K release have been conducted in row crop residues, there is a need for further research to evaluate K release dynamics from tree crop residues across climates and levels of rainfall and irrigation.

Many residue studies emphasize the strong relationship between initial K release and water application and that C:N does not strongly limit K solubilization [30,45,46]. However, some studies additionally suggest that K release rates in later stages may be influenced to a minor degree by plant structural components such as cellulose and lignin concentrations [71]. For instance, while legume green manures with very low C:N (around 9:1) may not show links between C content and K release [73], other green manures with higher C:Ns ranging from 10:1 to 30:1 showed a correlation between K release and hemicellulose and C content [72]. At higher (>30:1) C:N ratios, different types of nutshells contain varying concentrations of cellulose, hemicellulose, and lignin [39]. Rosolem et al. (2013) noted that while lignin may reduce the ability of water to enter plant tissues and solubilize K, biological degradation can help break cell barriers, promoting K movement [71]. Other plant compounds may influence K release; for instance, Hougni et al. suggested the waxy epicarp of cacao pod may require initial decomposition to enable K solubilization [43]. In that study, water saturation for 48 h resulted in only 11% K release from fresh cacao husks compared to 92% K release from partially decomposed husks. Future research focused on K supply from crop residues could assess and model all potential drivers of K solubilization, including water application rate and frequency, climate variables, C:N, C content and forms, decomposition rate, and microbial activity and community composition.

## 3. Potassium Availability

The bioavailability of solubilized K in the soil depends on several environmental and management-related factors. Soil K can be found in four functional pools: solution K, exchangeable K, non-exchangeable interlayer ("fixed") K, and structural lattice K in primary minerals [34,36,78]. Figure 1 provides a conceptual illustration of K dynamics in tree crop systems. Solution and exchangeable K are considered plant-available, while fixed and structural K are considered unavailable or extremely gradually available. These pools

exist in dynamic equilibrium and determine plant K availability [34]. Water application increases K availability in soil solution as K ions desorb from cation exchange sites and K solubilization occurs from organic, fertilizer, and mineral sources. While K can enter and exit mass flow streams, diffusion is the main mechanism of K movement in soil solution to roots [79]. Soils high in vermiculite, hydrous mica, or other K-fixing minerals can trap or fix K in interlayers [79–81]. This fixation can make large quantities of applied K unavailable and may be worsened by K-depletion from a history of intensive agriculture [33]. However, increased SOM content has been associated with reduced K fixation and improved K availability attributed to interlayer exchange and organic molecule adsorption sites [82,83]. Appropriate application of K-rich residues may assist in saturating K-fixing soils [33,34].

**Figure 1.** Conceptual diagram of the potassium cycle in tree crop systems. Processes are in italics and pools are bolded. In tree crop agroecosystems, K pools and processes overlap and interact more than pictured here. Potassium primarily moves through the soil by diffusion and can diffuse in and out of mass flow streams. The mechanism and rate of K movement depends on location in the soil and water dynamics. Potassium losses from leaching tend to be minimal unless in sandy soil [79]. Plant K can be stored in perennial organs, exported during crop removal, and recycled during litterfall and residue return.

While organic sources of K are capable of increasing solution, exchangeable, and fixed K, increases in soil exchangeable K (XK) are the most commonly reported due to the relevance of XK for changes in plant availability. Soil XK is often measured using ammonium acetate extraction and has been shown to increase in tree crop systems under cacao husk amendments [84,85], coffee husks [86], a mixture of coffee husks and pulp [29], pecan husk mulch [62], macadamia nutshells [87,88], bark mulch and alfalfa residues [40]. Composted olive pomace and olive mill wastewater can be applied to supply K and increase orchard soil XK, cation exchange capacity (CEC), SOM, N and other nutrients [31,89–92]. Citrus pulp residues have been shown to increase soil XK, other cations, and SOM content [93–95] and can be used as an alternative to expensive fertilizers [90,91]. In other crop systems, increased soil XK has been found under rice straw [34], green manures [66], cereal residues [96], cover crops [97], wheat straw [17], and a mixture of compost and wood scraps [55]. In one study, rice straw application significantly increased solution K, XK, and lattice K within the upper 60 cm in a sandy loam soil [34]. That study proposed that

increased solution K could be explained by direct K inputs, reduction in K fixation, and K solubilization and release due to interactions of SOM with clay.

However, the movement and fate of solubilized K depends on many factors such as minerology, CEC, pH, SOM, soil nutrient concentrations, water dynamics, environmental conditions, and soil management [25,36,42]. For instance, green manure residues led to higher and more immediate plant K uptake in coarse loamy soil compared to fine silty soil [32]. A soil trial evaluating pecan husk mulch found water extractable K was higher in sandy soil than finely textured soils after four weeks [62]. While K leaching through the soil profile can occur in sandy soils under high water application [36], surface-applied K is not commonly prone to leaching below the rootzone due to the inherent CEC of most soil minerals. Several studies indicated that K movement from surface crop residues was concentrated in the top 15 cm of soil [33,42]. The depth of K movement in the soil profile can be influenced by application rates and residue K content [42,71]. Supplementary Diagram 2 provides a template for calculating amendment application rate to achieve a given K input rate. As K movement relies on water availability, drought can restrict K diffusion rates while simultaneously limiting root growth [98]. Examining the movement of K across pools and evaluating processes of K fixation, release, and leaching beyond the root zone would greatly benefit K balance models [33]. A better understanding of the fate of K in agricultural soils would improve prediction methods for K availability to guide amendment management strategies [38].

## 4. Impacts of Increased Soil Health on Potassium Availability and Nutrient Cycling

Soil organic carbon (SOC) plays a critical role in sustaining agricultural systems by enhancing soil fertility and maintaining productivity [35]. As a critical functional component of soil health, SOC improves soil structure and water holding capacity while providing substrates for soil biota [55]. While conventional agricultural practices are known to deplete SOC, strategies to manage organic matter such as residue retention can reduce the severity of carbon losses. For instance, increased SOC has been found under surface-applied pecan husks [62], husks and pulp from coffee [29], and rice straw residues [35]. A three-year study using wood mulch applications found that SOC increased 23% on one site and 87% at a second site attributed to differences in initial SOC between locations, and SOC was the soil component most positively correlated with increased yield [55]. Coupling residue retention with reduced surface disturbance can provide further SOC building benefits. For instance, zero tillage with residue retention can reduce rapid oxidation of organic matter since residues are not mixed with soil, thus slowing decomposition and building residue-stored carbon [7]. Under organic management practices, composted olive pomace application can improve soil quality in olive groves and provide long-term benefits such as carbon storage and reduced erosion [91]. While residue transformations can contribute a fraction of dissolved organic carbon to building SOM in agricultural systems [18,99], a substantial amount of total SOC is typically derived from microorganisms and rhizodeposition [50]. Further research is needed to evaluate potential interacting drivers of increased SOC under different residues, including contributions from increased plant root growth, rhizosphere C processes, microbial dynamics, and organic carbon from plant residues [50].

Considering the physical component of soil health, crop residue amendments have the capacity to improve SOC and soil structure over time while providing a surface mulch that can enhance soil water and K availability. For instance, soil bulk density has been shown to decrease under almond shell amendments [10], hazelnut husks [100], cacao husks [54,101], olive mill waste [102], and alfalfa mulch [40]. Residue cover may alleviate negative effects of soil compaction [99,103]. Additionally, soil aggregate stability has been improved by application of pecan hulls and shells [62], pecan wood chips [104], coffee husks [105], and alfalfa mulch [40]. As a barrier on the soil surface, coffee husk application can protect against erosion, reduce runoff rates, and produce cleaner water flows [105]. Retaining crop residues and mulches is a soil conservation strategy that can help build SOM while reducing erosion [18,20,105,106]. In semi-arid regions, studies found reduced surface evaporation

under almond shell amendments [16], pistachio shells, pistachio hull-based compost, and olive pomace [107,108]. Crop residue mulches can reduce evaporation [18,33] and improve water use efficiency to produce more crop per drop [107,109]. Increased available water content has been found under pecan husk mulch [62], hazelnut husk compost [100], almond shells [16], and macadamia husks [110]. Residue applications can increase soil available water and irrigation use efficiency [18,103]. Water infiltration can be improved under coffee husks [105], maize and soybean residues [18], and alfalfa mulch [40]. Crop residues can moderate the effects of salt buildup by enabling salts to leach out of the top surface layer due to enhanced water infiltration, reduced evaporation, and increased soil moisture content [7,18]. Potential gains in water conservation are particularly high in semi-arid regions, given that water availability is expected to become increasingly unreliable in the future [13,107]. An immediate soil surface barrier coupled with long-term improvements in soil physical properties have the potential to substantially improve water use efficiency and uptake of K and other nutrients.

Tree crop residue application can increase SOM over time and improve soil chemical properties that govern the availability and retention of K and other nutrients. Improved SOM levels can generate new exchange sites, chelate and solubilize ions, and increase nutrient availability [108]. For instance, increased soil CEC under coffee husk amendments has been attributed to changes in SOC, pH, and decomposition by-products [29,86]. However, soil type may impact whether CEC changes under residues. For example, Carmo et al. (2016) found coffee husk applied at 4.7 Mg ha$^{-1}$ increased soil CEC in only one of three soil types [86]. In forest ecosystems, litterfall can contribute to replenishing cations and buffering soil acidity [111]. In addition to SOM exchange sites, plant residues themselves can absorb and adsorb K during decomposition, enabling K release and plant uptake later in the season [45]. Nutshells such as pecan, cacao, and almond have been characterized by high lignin content and high phenolic and carboxylic functional groups, which favor cation adsorption [6,112,113]. Further research is needed to investigate the dynamics of high C:N residue cation release and adsorption under different water regimes. In addition, decomposition processes release organic acids that can generate negative charges and preferentially adsorb divalent and trivalent cations, freeing up negatively charged sites on soil colloids that help retain K within the root zone [33,42,96]. This complexation has been shown to moderate high $Al^{3+}$ levels under residues [97]. While long-term, repeated residue applications at appropriate rates can build SOM and nutrient reserves [7], further research is needed to evaluate changes in SOM and CEC across orchard soils in tree crop agroecosystems.

Over time, improved SOM under tree crop residues can moderate soil pH toward neutral, promoting nutrient availability. Several studies recommended residues such as liming materials to increase pH [29,114]. For example, cacao husk compost may help raise pH and alleviate Al toxicity in soils where extended cacao production has reduced pH [84]. In another study, cacao husk amendments increased pH to 6.9 compared to the pH of 5.4 in control plots, which was accompanied by increases in available P, K, Ca, and Mg [101]. A lab incubation trial found higher soil pH under coffee husk amendments compared to the control after 330 days across three soil types [86]. Within three months, Kasongo et al. (2011) found coffee pulp and husk application increased soil pH, XK, Ca, and Mg while reducing Al toxicity [29]. After one year, soil pH increased from 5.21 to 6.14 under 5 tons/ha and 6.24 under 20 tons/ha; maximum CEC occurred under the highest application rate. Residues from certain species such as black oats can accelerate the mobility of surface-applied lime through topsoil layers likely due to complexation between organic ligands and divalent cations [96,97]. Conversely, other studies have indicated crop residues can ameliorate basic soils by reducing pH. For instance, almond shells in avocado orchards reduced alkaline soil pH, which increased available phosphorus after 10 years [10]. Since the almond shells contained 0.262 g kg$^{-1}$ P, researchers concluded pH reduction likely mobilized otherwise unavailable soil P. In another study, pistachio hull-based compost increased available P and Zn by lowering the pH of calcareous soils [108].

Future studies are needed to investigate the potential for crop residues to moderate pH levels from both acidic and alkaline extremes and associated effects on nutrient availability in agricultural soils.

High carbon residue application can increase soil microbial biomass and activity while shifting functional community composition. For instance, studies have found increased microbial biomass under macadamia husk mulch [110], rice straw residue [35], and bark mulch [40,115]. Similarly, soil microbial respiration has been found to increase under maize and soybean residues [106], leaf litter [115], and mixtures of compost and wood chips [55]. Residue C:N strongly impacts microbial growth and respiration as certain microbial groups are capable of utilizing high-C substrates more successfully than others [63]. As a result, high C:N amendments tend to promote more carbon efficient microbes such as fungi [40,116]. As materials degrade, distinctive shifts in community composition occur due to changes in available substrates, a phenomenon observed under almond shell amendments, for example [51,117]. However, some studies show conflicting results about whether low vs. high C:N amendments improve microbial community carbon use efficiency (CUE, proportion of $C_{assimilated}:C_{respired}$) likely due to site-specific environmental trait filtering [14]. However, a variety of C inputs with different C and N availabilities, including retained residues, could help maintain resources that support microbial CUE and avoid a shift to an overabundance of inefficient microbes [14]. In these ways, the integration of crop residues into soil management offers a strategy to help diversify soil carbon inputs, maintain more functionally diverse soil microbial communities, and maintain microbial CUE in agroecosystems.

More broadly, it is well-established that microbial biodiversity is essential for maintaining agricultural soil productivity and quality [35,118]. High amounts of taxa can impart functional redundancy and a "portfolio effect" that buffer microbial processes against environmental stressors [119]. Improved soil microbial diversity provides a greater variety of nutrient cycling functions. For instance, diverse species of ubiquitous soil fungi and bacteria are capable of solubilizing K from mineral sources, making K available for plant uptake and microbial use as the most common osmolyte in living cells [120–123]. While the response of microbial biodiversity to agricultural management is complex, conventionally managed soils often contain lower microbial species richness than organically managed or undisturbed soils [124–126]. Specific management approaches in organically managed systems can cause distinct shifts in microbial community guilds [125,126]. However, the direction of shifts in microbial species richness and biodiversity in response to management remains challenging to predict in the field due to variables such as climate, litter quality, SOC, N supply, vegetation, crop rotations, root exudates, soil pH, and impacts of climate change [35,117,127,128]. Nonetheless, agroecosystem management strategies are needed to mediate biotic homogenization in crop systems and improve functional trait diversity and associated ecosystem services, such as nutrient retention [129–131]. Residue application is one of many soil health building strategies that can provide carbon and nutrient inputs that support microbial community functioning and biodiversity.

## 5. Effects of Potassium-Rich Organic Matter Amendments on Crop Performance

Potassium uptake and transport are closely linked to water dynamics and fruit sink demands in tree crops [21,25,132]. As plants rely on available water for transport of K across membrane barriers, K uptake is strongly related to soil water content [21,64]. Mulches provide a physical barrier on the soil surface that improves temperature and soil moisture conditions for fine root growth, which can enhance nutrient uptake. Tree crop fine root growth has been shown to increase under almond shell mulch [16], macadamia husk mulch [88], and bark mulch [116]. Surface residues can increase available K in the upper soil layer where roots proliferate, and higher plant K uptake can facilitate root growth [7,33]. Root exudates assist in making K and other cations plant-available in soil solution through acidification and chemical reactions that can mobilize cations [114]. In addition, root exudates and decomposing residues can contribute labile carbon compounds

that stimulate microbial activity [61]. Since plants take up solubilized K, tree crops do not discriminate between different K sources to fulfill K demand [26]. Transpiration rates under amendments have been shown to increase compared to unamended controls [108], suggesting improved water uptake rates. Higher soil water can facilitate the movement of K into soil solution and improve K availability. However, effects of available water on plant K uptake may depend on factors such as site and cultivar. For instance, while Zipori et al. (2015) found no effect of irrigation level on olive leaf K concentration at one site, at another site leaf K concentrations increased with higher irrigation levels with substantial differences between cultivars [133]. Potassium is highly mobile throughout plant cells and tissues with primary functions as an osmolyte and counter-ion, and roles in enzyme activation and protein synthesis [64]. The timing of K demand is strongest during fruit development or nut fill, as shown in the example in Figure 2 below [25,132,134]. It plays a central role in carbohydrate transport in developing fruit and has been linked to fruit and nut quality [24,25,132]. Potassium plays an essential role in plant water dynamics, contributing directly to osmotic potential [64]. Sufficient K uptake is critical for growth, yield, and long-term plant health.

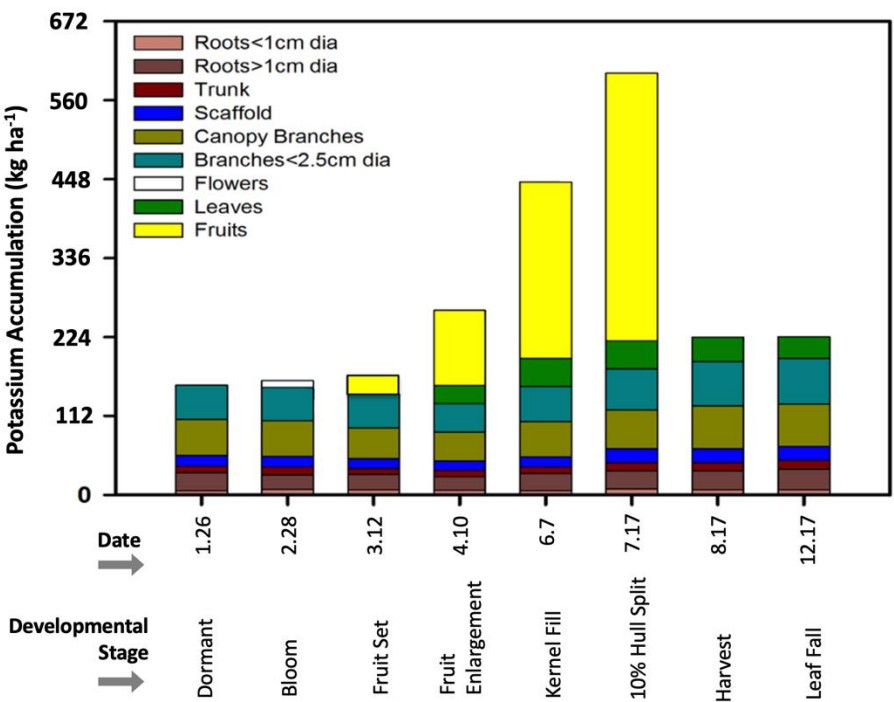

**Figure 2.** Potassium allocation in almond tree tissues, California, 2012. 83% of total K accumulated during a season was allocated to fruit tissues in almond trees. Adapted from Muhammad et al. (2020) [134].

Potassium application is an agronomic practice used to increase plant tolerance to temporary water shortages as it assists plants in responding to short-term water deficits [98,135]. Potassium is utilized in nearly all physiological plant processes that involve water, including stomatal regulation, assimilate translocation, and water transport. Optimizing K uptake can mitigate water stress through improved water use efficiency [98,136]. The roles of K in the physiological and molecular mechanisms of drought stress resistance include contributions to cell elongation, cell membrane stability, aquaporins and water uptake, osmotic adjustment, stomatal regulation, and detoxification of reactive oxygen species [137]. Potassium fertilization can increase hydraulic conductance of xylem and solute sap content [98]. Leaf K levels modulate the severity of the effects of water stress on photosynthesis through osmotic adjustment [64,138]. One study indicated K fertilization can induce isohydric

stomatal behavior, which increased responsive stomatal closure during water stress while increasing evapotranspiration in maize [136]. Across numerous crop types, studies have recommended ensuring sufficient K supply to alleviate the detrimental impacts of mild water deficit on plant growth, crop development, and yield during critical periods of crop K sensitivity [135,136]. Further research is needed to closely evaluate factors potentially influencing plant response to K from organic amendments, such as amendment application rate, existing K fertilization, water management, soil type, species and varietal differences. While plants may rely on sodium or small organic substances to maintain osmoticum under low K, meeting the minimal baseline K requirements for a given species is essential for optimal crop functioning [64]. Given projected global climate changes and the central role of K in water use efficiency, providing adequate K supply from an organic residue layer that simultaneously maintains soil water content could provide a critical strategy for maintaining crop productivity, particularly in arid and semi-arid agroecosystems.

Potassium deficiency can impede plant water dynamics, growth, metabolism, photosynthesis, carbohydrate transport, and resistance to stress. Stomata do not operate as efficiently under K deficiency, assimilate transport to roots is greatly reduced, and root growth slows [64]. As K-deficient plants tend to be more susceptible to damage caused by drought, maintaining adequate crop K is critical for plant drought resistance for many species [36,64,137]. Reduced K uptake can lower water uptake by reducing the activity of aquaporins [21]. Potassium deficiency can lead to reduced turgor, lower enzyme activation, metabolic disorders, and strong limitations on photosynthesis [64]. Insufficient K can accelerate premature leaf senescence and reduce numbers of flowers and fruits in subsequent years [23,139]. Increased susceptibility to both biotic and abiotic stressors occurs during K deficiency; this reduced stress resistance is attributed to increased ROS production [64]. At the crop system level, long-term soil fertility depletion from intensive agriculture can create nutrient imbalances as high quantities of nutrients are exported annually in harvested crops and crop residues [140]. When residues are not returned, a more severe net-negative K balance can develop in productive regions [34,45]. For instance, nutrient export in cacao beans and husks over time can remove substantial quantities of K and deplete soil K reserves unless replenished [67,69]. Cacao husks and composted processing waste amendments can reduce this net nutrient export and address K losses [43,67]. Potassium deficiency can develop due to insufficient K fertilizer use and increased yield demands, as well as innate soil K fixation and excessive N fertilizer use [25]. Nutrient retention tends to be low in soils that have been farmed for many decades. In addition, degraded soil structure and compaction can reduce K availability and uptake by reducing soil solution mobility [17,36]. The utilization of crop biomass wastes as amendment sources could be particularly useful in tropical regions with nutrient-depleted soils [60,101,140,141] and arid regions where water conservation is needed [16,62,107]. In regions where K fertilizer is less financially accessible, tree crops may be K-limited [142]. Tree crop growers may opt to supply nutrients from organic residues as a cost-effective alternative to ensure sufficient supply [142,143].

On occasions when studies have found increased yields under nutshell amendments, yield effects are most commonly attributed to increased soil water content and uptake of K and other nutrients. Residue mulches provide a physical barrier over the soil surface that can reduce soil evaporation, improve soil water storage, and lower tree water stress. This can promote transpiration, nutrient uptake and translocation, and carbon assimilation, which can directly increase biomass production. These improved water and nutrient dynamics have been linked with yield increases. For instance, a study in a water-limited region of Iran found fig trees under almond shell mulch produced higher quality fruit and higher yields while increasing leaf width, leaf number, shoot growth, and shoot diameter [16]. A trial in avocado found yield was maintained and occasionally increased under almond shell mulch, which mitigated drought conditions in Spain's Mediterranean climate [10]. In a study with young olive trees, pistachio shell mulch maintained less negative stem water status, increased stomatal conductance and chlorophyll fluorescence ratio [107]. Another study found composted pistachio hull and rice husk significantly increased shoot

and root K concentrations as well as shoot fresh weight [9]. Macadamia husk mulch can increase macadamia yield and foliar K [37,88]. Cacao pod husk amendments can enhance cacao seedling stem girth and height [143], as well as leaf K levels and growth parameters of cashew seedlings [85]. Similarly, coffee husks have been shown to improve cashew seedling development, increasing seedling leaf count, plant height, leaf area, biomass, and leaf K, N, P, K, Ca, and Mg [60]. Cacao pod husk used as an organic fertilizer can improve maize yields [53] and okra yields, root length, leaf Ca, pod weight and nutrients [101]. Solid olive mill waste application can increase olive fruit productive efficiency, dry weight, and yield over time [102,144].

Similarly, increased tree crop yields under wood mulch have also been attributed to improved water and nutrient dynamics. A study with wood chip mulch led to a 20–30% savings in irrigation water while improving apple tree growth (despite its high C:N) and prompting extensive fine root growth near the soil surface, indicating mulch likely improved conditions for tree roots [145]. Similarly, wood chip mulch led to exceptional tree growth in a study in sweet cherry, which researchers attributed to greater water availability [61]. In another study, apple trees under bark mulch had larger average trunk cross-sectional area three years after application [40]. Grinding and incorporating woody tree crop biomass into the soil led to higher yields, a 20% increase in irrigation water use efficiency during drought stress, and improved soil water retention and soil nutrients [12]. Neilsen et al. (2014) noted the tradition of using mulches to address K deficiency, in particular, in apple production systems [40]. A study in a semi-arid region found that K-rich alfalfa mulch in coarse-textured orchard soils helped address K fertilization issues under drip irrigation in apple orchards [146]. Mulches have been shown to improve soil physical properties, moderate water stress, increase trunk circumferences, tree size, leaf K concentrations, and yields in apple orchards in semi-arid regions [146,147].

Studies in row crop systems supported these findings and suggested that increased yields under residues likely result from improved soil physical conditions that promote water and nutrient availability and uptake [33]. Singh et al. (2018) found residue retention increased rice and maize K content, kept 75–80% of total K on site, and increased yields, which were attributed to improved soil physical conditions [33]. Mulching can significantly enhance yields and water and nutrient use efficiencies of maize and wheat [15,148]. Iqbal et al. (2010) found that wheat mulch and no-till increased soil K availability, delayed the onset of crop water stress in a semi-arid region, and improved root development, soil water utilization, and grain yield [17]. The authors suggested that mulch application may facilitate nutrient uptake by maintaining greater soil water content for longer time periods. Residue retention in a maize–wheat rotation has been shown to increase total chlorophyll, leaf area index, carotenoids, seedling establishment, nutrient uptake, root growth, grain yield, and protein content [7]. These improvements were attributed to enhanced aspects of soil nutrient cycling such as physical characteristics, nutrient availability, SOC, soil microbial biomass, and enzymatic activity.

Research from diverse crop systems has indicated that recycling crop residues as mulches can potentially contribute to closing yield gaps between attainable and actual yields, particularly in arid climates and low nutrient input systems [15]. However, delineating causal factors of yield differences is often complex due to numerous interactions between the residue, soil, crop, climate, microbial communities, fertilizer inputs, and other site-specific factors [46,106,147]. In some cases, crop residues may not lead to yield effects, or no clear relationship can be found between yield and effects of residues [87]. Additionally, while many studies have found increased soil XK under nutshell amendments and other crop residues, fewer studies have measured and reported leaf K status over time. For instance, a study with nutshell amendments found increases in soil XK but no differences in foliar K in macadamia [87]. A "dilution effect" of increased growth may be partially responsible for less frequent observations of increased leaf K levels [25]. Researchers have suggested that long-term field trials are needed to investigate the effects of residues on yield and crop physiology [106,147].

## 6. Future Research Directions: Building the Organic Layer in Orchards and Implementation Considerations

Nutrient cycling processes in forest and orchard agroecosystems often promote plant nutrient availability. Decomposition and nutrient release processes from plant residues play major roles in global carbon and nutrient cycles [63]. In forests, litter layers comprised of leaves, woody biomass, and nutshells supply notable amounts of nutrients such as N, Ca, and K [111]. In regions with low soil nutrients, forest ecosystem productivity is strongly influenced by nutrient cycling efficiency [111]. For instance, in a study in chestnut forests the annual return of Ca, Mg, and K through litterfall corresponded to 35% of the available soil pool of these cations [111]. In this study, leaves were richest in N and Ca while husks contained high N, K, and Ca. Similarly, leaf litter in tree crop systems can build an active N pool capable of net N mineralization [38,115]. Orchard leaf litter N can contribute to the balance of N mineralization and immobilization, which is influenced by N management history and orchard floor practices [149]. Leaf litter decomposition is a fundamental ecosystem process closely linked to nutrient supply for agroforestry tree species [44]. Mulching plays an important role in avocado production in California and creates a visible series of litter layers at different stages of decomposition [150]. While fresh litter layers with high C:N may immobilize soil N, lower, more decomposed litter layers typically have reduced C:N and release more N [150]. Understory tree crops such as cacao and coffee are typically grown under shade trees that provide substantial organic matter and nutrient inputs from leaf litter [67,151]. In coffee agroecosystems, nutrient losses can occur due to crop removal and long-term monocropping, while leaf fall, pruning, organic matter application, and intercropping can enhance soil nutrients [151,152]. These studies indicate that tree crop systems can be managed to optimize the inherent litter layer and integrate recycled nutrients into nutrient management strategies.

Orchard floor management plays an important role in organic tree crop production systems, given the central focus on managing SOM and more limited options for fertilizers and herbicides [56,146]. Some tree crop systems such as apple, macadamia, walnut, and almond utilize on-ground harvest practices, which typically require a bare orchard floor to enable crop pickup. This is accomplished through removal of organic litter and intensive herbicide use in conventional systems. On-ground harvest can lead to soil degradation, erosion, air pollution due to dust, and a longer time period for potential pest contamination [58,110,153]. In apple orchards, bare orchard floors are maintained by herbicides in conventional production systems and cultivation in organic systems, both of which have detrimental impacts on soil quality, SOC, N cycling, beneficial biota, and nutrient availability [57]. For instance, glyphosate applications reduced recycling of orchard floor vegetation and resulted in lower apple tree leaf K approaching deficiency, compared to mulch amendments [40]. Prolonged herbicide reliance in the tree row reduces plant biomass return, depletes SOM, and increases susceptibility to erosion [154]. In addition, recurring machinery passes and orchard floor disturbances can damage feeder roots responsible for nutrient and water uptake [56] and offset the benefits of organic matter amendments [155]. Alternative practices to on-ground harvesting are urgently needed to address these issues and improve nutrient management [153].

These tree crop management challenges highlight the need to further evaluate potential benefits of residue retention and reduced orchard floor disturbances on nutrient cycling, water dynamics, crop performance, air quality, and economic savings [56,57,153]. Organic matter retention and reduced disturbance can be integrated with other soil health building practices such as cover cropping with legumes, which are capable of lowering nutrient losses and increasing N stored in tree biomass [57,61,156]. In deciduous tree crops, substantial amounts of nutrients and carbon are returned to the soil annually through leaf abscission, mowing of vegetation, rhizodeposition, and tree pruning, which are processes that can be deliberately managed to enhance nutrient cycling [38]. At a site with a history of mulch applications, one study found net N release from eucalyptus mulch (C:N 51:1) doubled after three years [150]. Nutshell applications can contribute to the formation of

new organic layers in avocado orchards over time [10]. Returning higher rates of plant litter while reducing soil aggregate disruption can mitigate agricultural soil carbon depletion [154]. Slowly building SOC content can provide long-term nutrient cycling benefits such as fertilizer N retention and supply to tree crops [149]. High C:N amendments in particular may help build long-term soil N reserves [57]. Some research points to the process of microbial N immobilization under high C:N amendments as a potential tool to mitigate nitrate leaching potential, reduce denitrification, and improve N cycling without limiting crop-available N [12,13]. Evaluating nutrient release dynamics from high C:N residues will improve predictions of nutrient availability for tree crop uptake to guide management [38,44]. Site-specific orchard floor management strategies can be tailored to optimize nutrient and water cycling and crop health as trees mature over time [57,61]. This body of research highlights great potential to develop our understanding of nutrient cycling and availability across trophic levels under reduced orchard floor disturbance and organic residue retention.

However, tree crop growers may face a variety of practical constraints potentially limiting the implementation of nutshells and other high C:N residues as organic matter amendments. Contextual factors such as access to residues, application equipment, labor, and pest and disease considerations may present different implementation challenges across crop systems and regions. Evaluating factors that promote implementation and potential barriers to adoption is essential for future applied research and communications focused on organic matter amendment use [157,158]. Further investigation of social, economic, and regional constraints will enable more effective and holistic agricultural recommendations that serve tree crop growers and the public [67]. Multidisciplinary research examining these questions will assist growers of diverse scales and management approaches in adjusting tactics to support system efficiency and profitability. For instance, current literature suggests that the application of composted cacao and coffee pericarps could provide key agroecosystem services such as K cycling and pathogen suppression. Management considerations in these two tree crop systems provide examples of specific benefits and constraints for further evaluation.

Composted cacao pod husk amendments can be used to provide K, enhance tree nutrition, and suppress plant pathogens [43,84]. These ecosystem services could provide meaningful benefits, given that cacao productivity is often limited by soil fertility, pest pressures, and post-harvest practices [67,159]. While cacao husks are well-known to be nutrient-rich, piles of husks scattered on cacao farms can cause sanitation issues and amplify pest pressures [140,143]. Effective composting practices can kill certain pests such as cacao pod borer larvae and black pod disease (*Phytophthora palmivora*) [69,140] and suppress mycelial growth of *Phytophthora megakarya* pod rot [84] although it may not eliminate viral diseases [142]. Additionally, compost applications may induce systemic plant defenses against diseases by enhancing the growth of beneficial soil microbial consortia [84]. Simultaneously, composting can improve cacao husk amendment pH and increase nutrient concentrations [140]. However, further research is needed to evaluate implementation considerations, constraints, and which specific insect pests and plant pathogens could be addressed through composting [85]. Cacao management tends to be labor-intensive, and logistics of husk transportation and application may pose barriers to implementation [85]. Labor shortages and fluctuating market values may limit the adoption of best management practices in cacao [67,159]. However, a study in Nigeria found that farmers using cacao pod husk as fertilizer gained triple the profit per hectare than farmers not using this amendment [160,161]. To optimize labor efficiency in resource-constrained smallholdings, rotating pod breaking stations and sequential mulching in small field areas could lower labor requirements. Considering that cacao yields are often limited by fertility and disease in many regions [69], improved husk management could offer an avenue for lifting yield limits if logistical constraints are adequately addressed.

Similarly, composting coffee residues can be used to improve amendment pH, increase K content, suppress certain pathogens, and benefit crop performance. While initial

coffee wastes can be acidic, composting husks together with plant and animal wastes can dramatically improve pH while increasing nutrient concentrations and building beneficial microbial communities [60,162,163]. For instance, composting coffee husks with manure and beneficial microbial inoculants has been shown to enhance pathogen suppression of *Rhizoctonia solani* while increasing pH and nutrient content [164]. Coffee husk amendments have been shown to improve soil K, N, and C, fertility, and yield while reducing pollution and erosion due to runoff [29,60,86,105]. While different types of post-harvest processing methods affect K content, high K release across many coffee residue types indicates residues can substitute for mineral K sources [68]. However, the implementation of sustainable practices in coffee varies widely across regions depending on factors such as farm size, external input use, mechanization, economic stability [165,166]. Regular access to substantial amounts of organic matter as nutrient inputs in organic coffee production can be challenging for smallholders [151]. Typically, coffee is often processed offsite and residues might not be easily transported back to coffee farms, which are often located on steep slopes at high altitudes [151]. Access, labor, transportation, and farm financing are likely limitations for coffee producers interested in applying coffee husk amendments. Further studies are needed to evaluate the practical constraints potentially limiting the adoption of this practice in addition to nutrient supply dynamics and the potential to enhance disease suppression.

## 7. Conclusions

In summary, a growing body of research points to the substantial potential of regional crop residues to be recycled as soil amendments in tree crop agroecosystems. Relatively high C:N amendments can supply K and other nutrients, promote many components of soil health, and enhance crop water use and crop performance. Current literature has established that water application is the central driver of the solubilization of K ions from plant residues into soil solution. Evidence from tree crops, other permanent crops, field crops, and forest ecosystem studies indicates that residue retention can be integrated with soil management practices to provide plant and soil benefits. Further research is needed to assess all potential factors influencing K release, K solubilization rates, K fate, crop K and water uptake, impacts on plant function and yield across crop types, management approaches, and regions. Findings indicate great potential for recycled residue K to supplement or substitute for fertilizer K in tree crop systems. Impacts could be particularly meaningful in areas where agriculture has depleted soil K and SOM and where water may be a limiting factor. Additionally, future studies could evaluate associated N dynamics, effects of complementary orchard floor management practices, and potential use within integrated pest management approaches. Further investigation of contextual constraints to adoption are essential, including access to residues, transportation, labor, and local socioeconomic considerations. Interdisciplinary research is needed in order to fully understand the likelihood of grower adoption and to support management recommendations that are deliberately tailored to unique agroecosystem contexts.

**Supplementary Materials:** The following are available online at https://www.mdpi.com/article/10.3390/agriculture11070580/s1. Supplementary Diagram 1: Tree Crop Pericarp Examples: almond, cacao, coffee. Supplementary Diagram 2: Example Application Rate Calculation—Almond Hulls and Shells.

**Author Contributions:** Conceptualization, P.H.B., S.D.S.K., E.M.A., S.K.; methodology, E.M.A.; investigation, E.M.A., S.K., E.E.S.; resources, E.M.A., S.K., E.E.S.; content curation, E.M.A.; writing—original draft preparation, E.M.A.; writing—review and editing, E.M.A., S.K., E.E.S., P.H.B., S.D.S.K.; visualization, E.M.A., S.K.; supervision, S.D.S.K., P.H.B.; project administration, S.D.S.K., P.H.B.; funding acquisition, E.M.A., S.D.S.K., P.H.B. All authors have read and agreed to the published version of the manuscript.

**Funding:** This research was funded by the Western Sustainable Agriculture Research and Education Grant, grant number SW20-912, and the Foundation for Food and Agriculture Research, grant number CA18-SS-0000000260.

**Acknowledgments:** Authors would like to thank W.R. Horwath and D. Geisseler for feedback on Figure 1.

**Conflicts of Interest:** The authors declare no conflict of interest.

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
