# Peer review of "A Review of Potassium-Rich Crop Residues Used as Organic Matter Amendments in Tree Crop Agroecosystems"

_agriculture, doi:10.3390/agriculture11070580_

Round 1

Reviewer 1 Report

In their review paper, Andrews et al. discuss the implications and benefits of organic residue amendments focusing on potassium dynamics in soils and fruit trees. The authors specifically highlight the potential of this management strategy to recycle nutrients and its many positive aspects in plant performance and environmental integrity. I therefore find this review is highly relevant for readers of “Agriculture” journal.   

Overall, the MS is well-written, comprehensive and fluent. Few minor typos and suggestion can be found in the attached PDF file.

I have two mine concerns that I suggest addressing in the revised version:

  1. The positive approach to soil amendments is understandable, but in some cases more sceptic and careful approach should be taken. For example: while many studies presented here show positive effect of K on yield, just as many showed no effect. Similarly, the positive effect of K on abiotic stress is far from being universal phenomena, many studies show lack of response and substitution of K by sugars or Na. These reservations should be discussed as well. More examples are listed in the MS.
  2. Some of the text is repetitious as can be reduced significantly. For example, large section (# 6) is dedicated to N cycle while from the title and abstract N does not seems to be an objective of this review? In the “future directions” section much of the information was already discussed earlier. I feel section 6, 7 and 8 can be combined into single short section.

Overall

Reviewer 2 Report

Potassium (K) is one of the essential nutrients in plants and in recent years is one of three (including nitrogen and phosphorus) that are commonly in sufficiently short supply in the soil to limit crop yields on many soil types. Therefore the subject of this paper is interesting and fulfils the scope of AGRICULTURE, MDPI Journal but some information needs to be completed.

Generally, the paper is well prepared and provides key information on the soil K cycle related to the use of various types of organic amendments. The manuscript contains a rich literature review (most of it not older than 10 years), which is its significant advantage. The paper division into appropriate chapters is quite clear, but I think that sections 6, 7, 8 should be combined and divided into appropriate subsections. In addition, authors must ensure that all applied abbreviations (SOM, SOC, C: N and others) have been properly explained.

Moreover, I suggest adding a small section on the potential of K losses/deficiency in soil and preventing them in agriculture as well as methods/techniques of assessing its content in soils and soil organic amendments.

Round 2

Reviewer 2 Report

Dear Authors,

I am fully satisfied with the responses on my sugesstions and all changes introduced to the revised version of yours manuscript.